# Effect of Landscape Composition and Invasive Plants on Pollination Networks of Smallholder Orchards in Northeastern Thailand

**DOI:** 10.3390/plants11151976

**Published:** 2022-07-29

**Authors:** Pattraporn Simla, Thotsapol Chaianunporn, Wangworn Sankamethawee, Alice C. Hughes, Tuanjit Sritongchuay

**Affiliations:** 1Landscape Ecology Group, Center for Integrative Conservation, Xishuangbanna Tropical Botanical Garden, Chinese Academy of Sciences, Menglun, Mengla 666303, China; pattraporn1916@gmail.com; 2International College, University of Chinese Academy of Sciences, Beijing 100049, China; 3Department of Environmental Science, Faculty of Science, Khon Kaen University, Khon Kaen 40002, Thailand; thotsapol@kku.ac.th (T.C.); wangsa@kku.ac.th (W.S.); 4Southeast Asia Biodiversity Research Institute, Chinese Academy of Sciences, Yezin, Nay Pyi Taw 05282, Myanmar; 5University of Chinese Academy of Sciences, Beijing 100049, China; 6School of Biological Sciences, University of Hong Kong, Pok Fu Lam Road, Hong Kong, China; 7UFZ-Helmholthz Centre for Environmental Research, Department of Computational Landscape Ecology, Permoserstr. 15, 04318 Leipzig, Germany

**Keywords:** biological invasions, invasive plants, land-use change, pollinator communities, pollination networks

## Abstract

Destruction of natural habitat, land-use changes and biological invasion are some of the major threats to biodiversity. Both habitat alteration and biological invasions can have impacts on pollinator communities and pollination network structures. This study aims to examine the effect of an invasive plant, praxelis (*Praxelis clematidea*; Asteraceae), and land-use types on pollinator communities and the structure of pollination networks. We conducted the study in smallholder orchards which are either invaded or non-invaded by *P. clematidea*. We estimated the pollinator richness, visitation rates, and pollinator diversity and evaluated the network structures from 18 smallholder orchards in Northeastern Thailand. The effect of landscape structure in the vicinity of the orchards was investigated, with the proportion of agricultural, forest, and urban landscape within a 3 km radius analyzed. The invasive species and land-use disturbance influence the pollinator communities and pollination network structure at species level was affected by the presence of *P. clematidea*. Bees were the most important pollinator group for pollinator communities and pollination networks of both invaded or non-invaded plots, as bees are a generalist species, they provide the coherence of both the network and its own module. The urban landscape had a strong negative influence on pollinator richness, while the proportions of agriculture and forest landscape positively affected the pollinator community.

## 1. Introduction

Destruction and the alteration of natural areas, in addition to biological invasions, are some of the major threats to biodiversity [1,2]. The isolation of habitat patches causes slower immigration by new species leading to invasion by non-native species. Moreover, humans drive the spread of alien species across geographical barriers, and between regions [3]. For example, international trade via new roads and railways can enable the spread of weeds and invasive species [4]. Invasive plant species can spread and become dominant species in communities, which has negative effects on native species and can change ecosystem functioning [5,6,7]. The integration of invasive plants into communities can negatively influence competition between plants as they share some floral traits that affect the pollination success of native flowering plants [8,9].

The consequences of invasion for pollinators and plant–pollinator interactions are context specific. Following the invasion of alien plant species, pollination networks lost specialist flower-visitor species leading to low network modularity, indicating a low complexity [10,11]. In addition, the establishment of invasive plants is likely to change the dynamics of the network by increasing the number of links and species persistence in pollination networks [12,13]. Nevertheless, both plant richness and pollinator richness were positively associated with landscape management. Landscape heterogeneity can maintain the resources for foraging and nesting and promote easier flow of pollinator individuals. Decreasing habitat availability can isolate populations of pollinators and directly affects species survival and interactions at the community level, such as the pollination network structure [14,15,16]. The consequences of increased pollinator species abundance are crucial to the stability of mutualistic networks [14,17]. When the landscapes become fragmented, the abundance and diversity of pollinators changes, which affects plant–pollinator interactions [18,19]. Thus, resource tracking is one of adaptive strategies employed to cope with spatiotemporal environmental dynamics for an ecological network [19,20,21].

However, to date few empirical studies have been conducted on the combined effect of landscape and invasion on plant–pollinator interactions. In Southeast Asia, the knowledge of biological invasion effects is relatively localized [22]. However, what remains even less clear is whether the invasive species may act in synergy with disturbances due to human activity to alter plant and pollinator communities on a regional scale. Additionally, biological invasions are poorly understood from the perspective of complex ecological networks [23]. Thus, understanding the causes and consequences of landscape variation with the presence of invasive plant species is important for answering fundamental ecological questions on plant–pollinator interactions, as well as for the application of conservation and landscape management strategies. Super-generalist invasions tend to change the core of the nested matrix and may increase the overall of nestedness of the networks, this may increase extinction risk of specialist species [24]. Ecological networks of species and their interactions can simplify the description and understanding with reference to various species groups [25,26].

Here we focus on the modularity of an entire pollination network and describe species roles with aggregated sets of interacting species as modules. For example, naturalized invasive plants have dominant roles (e.g., core or hub species) or are highly connected within the network. We focus on the invasive plant *Praxelis clematidea*, an invasive species which grows in dense stands enabling it to outcompete native flora and alter habitats [27]. Previous studies revealed its effects on soil nutrients and herbaceous plant diversity [28,29]. The presence of exotic plants can affect the pollination success of native plants if they receive high visitation rates from native pollinators. However, there is little information available of the effects of this species on different trophic levels and native plant and insect communities, despite its high invasibility across Asia [2,27]. Therefore, in this study we aim to identify the role of *P. clematidea* in pollination networks and examine the effects of the presence of *P. clematidea* and land-use types on pollinator communities (flower-visitor diversity, abundance, and richness), and the structure of pollination networks. Considering that *P. clematidea* may play an important role in the community, it is expected that the invaded areas may have a high visitor abundance and richness that may impact on plant–pollinator interactions, and show a high species interacted within the module. We also hypothesized that the effect of the invasive plants on pollinator communities may be greater in areas surrounded by urban area coverage. As the anthropogenic habitat commonly favors alien species distribution via vegetation management and changes in community-level plant phenology [30], they may influence the composition of the pollinator community [11,31], which may reflect pollination network structures.

## 2. Results

### 2.1. Plant and Pollinator Community

The 18 study sites contained a total of 134 flowering plant species in 58 families. Insect species observed interacting with flower-visitors (hereafter called pollinators) visited 41 flowering plant species of five families: 15 species of Fabaceae, 8 of Solanaceae, 6 of Apocynaceae, 6 of Rubiaceae, and 6 of Rutaceae. In this study, the peak of the flowering season occurred between February and May. The flowering period of *P. clematidea* was from the end of January to May, which overlapped with flowering of most plant species within the study sites, including the most abundant crops, such as longan (*Dimocarpus longan* Lour., in this region the flowering period is from February to April) and mango (*Mangifera indica* Linn., flowering period from January to Mar). The frequency of pollinator visitations varied in each month observed. There was no significant difference in any parameters of plant community composition between invaded and non-invaded sites (Table 1).

We recorded 177 morphospecies of flower-visitors from six orders of insects (class Insecta). The most common order was Hymenoptera (80 morphotypes), followed by Diptera (47), Lepidoptera (33), and others (Coleoptera 9, Hemiptera 7, Odonata 1). The majority of flower-visits were carried out by hymenopterans (71.84% of all species), followed by dipterans (21.81%), lepidopterans (5.41%), and other arthropods (0.94%). Hymenopterans had the most recorded visits overall, with 72% of all visits, followed by dipterans (22%), lepidopterans (5%) and others with 1% (Figure 1). Honeybees *Apis florea* and *Apis cerana* were the most common pollinators with 27.67% of all visits, followed by flies (21.83%), wild bees in the family Halictidae (11.86%), and stingless bees (7.07%).

Overall, pollinator diversity was not significantly different between the invaded and non-invaded sites (with 33% of species overlapping) but they differed in terms of pollinator groups (Figure 2). At the invaded sites, the most common pollinators were *A. cerana* with 26.87% of all visits, followed by flies 23.10%, and wild bees in the family Halictidae 8.96%. For the non-invaded sites, flies were the most common pollinator with 20.65%, followed by *A. florea* (16.21%), and halictid wild bees (14.07%). We found no significant differences in any pollinator community parameter between invaded and non-invaded sites (Table 1).

### 2.2. Pollination Network Structures

The pollination networks of all study sites were diverse with 177 pollinators visiting 73 plant species across the entire season, including 517 interactions over 18 study sites (with 26 plant species and 55 pollinators only found in the invaded network and 18 plant species and 63 pollinators exclusive to the non-invaded network, Figure 3a). In the non-invaded network, we recorded 115 pollinators visiting 38 plant species, with 281 interactions (Figure 3b), and the invaded network included 113 pollinator species visiting 49 plant species, with 283 interactions (Figure 3c). We found no significant difference in any network parameters (Table 2).

### 2.3. Network Modularity and Pollinator Roles

The standardized modularity value was significantly different between the invaded and non-invaded networks (*Q* = 0.59, *z* = 72.76, Appendix A). There were no differences in the participation coefficients (*c*) or within-module degree (*z*) values when compared in the same pollinator group between invaded and non-invaded network (Appendix A).

In the invaded network, the participation coefficients varied significantly between hymenopterans and dipterans (*p* = 0.035, Appendix A), and the within-module degree between hymenopterans and dipterans varied slightly, but statistically insignificant (*p* = 0.053). Meanwhile, in the non-invaded network, the within-module degree varied significantly between hymenopterans and butterflies (*p* = 0.022, Appendix A).

Once modules have been defined, we can classify the species’ importance to its own network (Appendix A). In the invaded network, the super-generalist pollinators (network hubs) were two wild bee species, and included *halictid bee* sp.2 and *Tetragonilla* sp.2, and one species of wasp-mimicking hoverfly *Stomorhina* sp. While two other bee species, *Amegilla* sp.1 and *Xylocopa aestuans*, acted as module hubs with highly connected species linked to many species within their own module. In the case of the non-invaded networks, the super-generalist pollinators were the honeybee *A. florea*, wild bee *halictid bee* sp.2, wasp-mimicking hoverfly *Stomorhina* sp., and ant sp.3. Module hubs included *halictid bee* sp.3 and *Ceratina* sp. As for the invasive plant *P. clematidea*, it was a peripheral species in the invaded network (*c* = 0.39, *z* = 0.09) with a low participation coefficient (*c*) and had few linked species within their module.

### 2.4. Effects of Invasion and Landscape Proportions on Pollinator Richness and Visitation Rates

The generalized linear model (GLM) analysis shows that total pollinator richness in the smallholder orchards was positively affected by proportion of forest (GLM; *z* = 3.406, *p* = 0.0007), and positively affected by surrounding agriculture landscape within a 3 km radius of each site (GLM; *z* = 3.498, *p* = 0.0005), and there was a significant interaction between plant richness and agricultural proportion (GLM; *z* = 3.087, *p* = 0.002). The proportion of urban landscape was negative related to the total pollinator richness (GLM; *z* = −3.580, *p* = 0.0003), and was significantly affected by the interactions between flower abundance and urban proportion (GLM; *z* = −3.217, *p* = 0.001), as well as being significantly affected by the interactions between plant richness and urban proportions (GLM; *z* = 3.087, *p* = 0.026).

The species richness of Hymenoptera was positively affected by the proportion of forest (GLM; *z* = 2.044, *p* = 0.04) and agriculture landscape (GLM; *z* = 3.371, *p* = 0.0007), but negatively affected by the proportion of urban landscape (GLM; *z* = −2.129, *p* = 0.033). Further, Hymenoptera richness was significantly affected by the interaction between plant richness and agriculture proportions (GLM; *z* = 3.307, *p* = 0.0009) and was a significant affected by the interaction between plant richness and forest proportions (GLM; *z* = 2.105, *p* = 0.035). Lepidoptera richness was negatively affected by the proportion of forest landscape (GLM; *z* = −2.222, *p* = 0.026) and there was a significant interaction between flower abundance and agriculture proportions (GLM; *z* = 2.012, *p* = 0.044). Further, Lepidoptera richness was significantly affected by the interaction between flower abundance and urban proportions (GLM; *z* = 2.086, *p* = 0.037). The proportions of each three landscape types within a 3 km radius did not, however, have an effect on Diptera richness (all *p* > 0.10, Figure 4).

For the visitation rate, the effects of landscape proportions depended on pollinator groups. The total visitation rate was positively affected by the proportion of forest landscape (GLM; z = 4.779, *p* = 0.04) within a 3 km radius of each site. The visitation rate of Hymenoptera was also positively affected by the proportion of forest landscape (GLM; *z* = 4.904, *p* = 0.039). The visitation rate of Diptera was, however, positively affected by forest (GLM; *z* = 8.781, *p* = 0.013) and agriculture landscape (GLM; *z* = 2.831, *p* = 0.018). Diptera visitation rate was further significantly affected by interactions between plant richness and forest proportions (GLM; *z* = 9.997, *p* = 0.01), and the interaction between flower abundance and forest proportions (GLM; *z* = 9.940, *p* = 0.01). However, we found no significant difference in the visitation rate of Lepidoptera compared to any landscape proportions within a 3 km radius of each site (all *p* > 0.10, Figure 5).

The effect of agricultural proportions was, however, different between the invaded and non-invaded sites. There was a negative relationship between agriculture proportions and the Diptera visitation rate in the invaded site and a positive relationship in non-invaded site (GLM; *z* = 3.115, *p* = 0.011). Furthermore, there was a significant interaction between invasion and forest proportions. The effect of forest proportions on the total pollinator richness, Hymenoptera richness, and Diptera visitation rates was different between the invaded and non-invaded sites. For total pollinator richness, this effect was positive in the invaded sites and negative in non-invaded site (GLM; *z* = 2.099, *p* = 0.036). The effect of forest proportions on Hymenoptera richness was positive in the invaded sites and negative in the non-invaded sites (GLM; *z* = 3.229, *p* = 0.001). The effect of forest proportions on the Diptera visitation rate was positive in the invaded sites, while in the non-invaded sites it was negative (GLM; *z* = 5.076, *p* = 0.037). In addition, the effect of urban proportion on total pollinator richness and Hymenoptera richness was different between the invaded and non-invaded sites. There was a positive effect on total pollinator richness in the invaded sites, while there was a negative effect in the non-invaded sites (GLM; *z* = −2.309, *p* = 0.02). There was also a positive effect on Hymenoptera richness in the invaded sites and a negative effect in the non-invaded sites (GLM; *z* = −3.242, *p* = 0.001). All statistics are available in the Appendix A.

## 3. Discussion

This study highlights the patterns of how invasive species influence pollinator communities and pollination networks and the importance of the landscape connectivity to sustaining pollinator communities. To our knowledge, this is the first study to investigate the impact of invasive species together with the synergistic effect of land-use on plants and pollinators in smallholder orchards with a particular focus on their interactions and the resulting structure of pollination networks. We investigated pollinator communities and the pollination network structure between invaded and non-invaded smallholder orchards. We also explored the role of *P. clematidea* on the pollination network. There are two important results: firstly, invasive species and land-use disturbance influence the pollinator communities. Second, the pollination network structure at species level was affected by the presence of *P. clematidea* and environmental factors. We will discuss these two findings in turn, ending with some thoughts on the implications for agriculture and pollination conservation.

### 3.1. Effect of Invasion and Surrounding Landscape on Pollinator Richness and Visitation Rates

The majority of pollinators differed between orchard types. The dominant pollinator of the invaded study sites was the Asian honeybee *A. cerana*, whereas flies were the most common pollinators of non-invaded sites. This study revealed the response of bees and flies to the invasive flower that had an effect on pollinator community composition, and may promote generalist pollinators within the study sites. However, the allocation to natural and cultivated habitats at a landscape level could influence the functioning of both of the invaded and non-invaded communities [14,32,33]. We found that the interaction between landscape proportions and plant community was influenced by the pollinator community. Vegetation attributes such as plant species richness and flower abundance are also important for insect foraging due to the morphology of flowers and pollinators [34,35,36].

The combination of biological invasion and landscape effects may impact the composition of the pollinator community. Furthermore, different pollinator groups’ responses to different environmental factors depend on their requirements [34]. The total pollinator richness and bee richness was influenced by urbanization, and the pollinator community is also associated with plant richness and flower abundance. Agriculture and forest landscape surrounding smallholder orchards facilitate the increase in bee richness and bee visitation rate. While a high urban proportion decreased bee richness, bee richness increased in the invaded site in the presence of *P. clematidea*. Consequently, the landscape effect suggested that the anthropogenic habitats commonly favor alien species distributions and facilitate generalist pollinators in the invaded area. As bees are dominant pollinators in the invaded site, an increase in bee richness in urbanized areas with *P. clematidea* present suggests that invasive plants can serve as pollination resources in any habitat [6,11].

In this study, we found the visitation rate of flies increased with a high proportion of agriculture and forest landscape. The visitation rate of flies is also influenced by plant richness and flower abundance. Fly visitation decreased in invaded sites surrounded by a high proportion of agricultural landscape, whereas invaded sites surrounded by a high proportion of forest landscape increased the visitation of flies. Because the response of native pollinators to plant invasion varies depending on the identity and characteristics of the invasive plant [21], insect competition for resources may result in a decrease in visits. While the presence of *P. clematidea* increased bee richness in the invaded sites surrounded by a high proportion of agricultural landscape, the visitation rate of flies decreased. Previous studies suggest that pollinators use a broader range of plants in the invaded site and resource levels can be compared based on flower species preferences [11,36]. The foraging behavior of generalist pollinators, such as honeybees, may force other pollinators to shift to less visited plant species [21,36,37].

Conversely, butterfly richness decreases with a high proportion of forest landscape surrounding smallholder orchards. The agricultural and urban landscapes, as well as flower abundance, influences butterfly richness. Results from previous studies revealed that landscape transformation in the tropics can provide extra resources for flower-generalist butterflies [38,39]. The relationships of flies and butterflies with habitat are more complex, with more specialist and specificity in flower choices. Bees need nectar and pollen from flowering plants, however, the fact that butterflies and flies lay eggs on host plants means associations are more complex as they require more than nectar and pollen from flowers [34,40]. Furthermore, characteristics of habitat with intermediate disturbance regimes may offer more resources for insects and provide the diversity of efficient pollinators [20,41]. Thus, the consequences of biological invasion by flower attractiveness and land-use influences pollinator richness within the community, regarding flower resources and habitat availability. Together, combined with the plant community and resource availability, the heterogeneous environment that surrounds smallholder orchards may offer a habitat that is friendly to pollinators [42,43,44].

### 3.2. Effect of Invasive Plant and Surrounding Landscape on Pollination Networks

In this study, we found no significant differences between invaded and non-invaded sites in terms of network-level structure. Contrary to our expectations, the invaded network was not obviously impacted. We found no shift in the specialization of plant–pollinator interactions, and the specialization was similar within both invaded and non-invaded networks. Previous studies found that invasive plants integrate into native plant–pollinator networks, based on the establishment of interactions [13,42,45]. This may be a consequence of the role of *P. clematidea* as a peripheral species in the pollination network that was visited by a lower diversity of pollinators and had fewer interactions in comparison to native plants. Previous studies suggest that invasive plant species are specialized and have lower species strength (sum of interaction strengths of the plants on a specific pollinator) when compared to native plant species [9,12,46].

Our results showed that pollinator species were significantly associated with the standardized modularity of the network, as the metrics of mutualistic networks are a function of the connections between the two levels (plants and pollinators) of a network and thus fundamentally affected by the generality and richness of network species [15,16,47,48]. Participation coefficients and the within-module degree of pollinator groups were not different between networks, but the contributions to a modular structure of each group differed within their network. The high participation coefficients of hymenopterans were a key predictor of differences in the modularity contributions within the network. The structure of the pollination network is associated with a high honeybee richness, as we found *A. florea* played the role of “network hub” that connected with the species in its own module and the entirety of the non-invaded network. Previous studies found that honeybees are sometimes not the most efficient pollinators when compared to wild bees [49,50]. This supports our finding that wild bees are high abundance pollinators in the invaded network, although the invaded sites received a high abundance of *A. cerana*. Despite not being the most efficient pollinator, *A. cerana* are also important as a connector with a high participation coefficient species and which connects different modules in the invaded network together. Therefore, invasive *P. clematidea* may not impact the network structure directly, but its presence affects the interchange of the pollinator community composition, which revealed that species levels differed between invaded and non-invaded sites (Appendix A).

In addition, the landscape characteristics were related to the pollination network structures [16,51,52]. The proportion of agriculture and forest landscape surrounding smallholder orchards is associated with increasing pollinator richness, which is reflected in the plant–pollinator interactions in the networks. Previous research had revealed that proximity to forest areas affects pollinator richness [52] and highly agricultural landscape favors the persistence of generalist pollinators [16]. The distribution of invasive plants in urban settings is also facilitated by urbanization, which has the potential to alter the pollinator community and composition of wild pollinators [11,31]. The richness of pollinators in the orchards is positively influenced by the connection of varied landscapes. Consequences of the combined effect of urbanization and biological invasion may alter networks of plants and insects. Hence, the increased generality of pollinators and the abundance of resources has positive effects on network stability.

### 3.3. Implications for Agriculture-Conservation and Future Research

Our study demonstrates that the proportion of the urban landscape influences the insect community within smallholder orchards and could negatively affect pollination services to crops. The reproductive success of crops is often dependent on insect pollinators [34,53]. Studies of plant–pollinator networks can support conservation strategies, and the coexistence of pollinators and plants within natural and agricultural systems [19,54]. Both agricultural diversity and the appropriate arrangement for invasive plants are associated with increasing pollinator dependence which could support pollination networks and provide resources for generalist pollinators that facilitate the pollination of native plants [55,56,57]. We found evidence not only that the richness of local vegetation influences the pollinator diversity, but also that invasive flowering plants can actually increase insect diversity. As flowering period of many crop species are limited, and other floral resources are needed to sustain pollinators outside these peak flowering periods. Once invasive plant species expand their ranges, they may alter phenology and can actually help maintain pollinators when few floral resources are available [57,58,59]. We suggest further studies should investigate the effective competition for pollination between the invasive plant *P. clematidea* and native plants. Identifying the effectiveness of pollinators with invasion and sensitive native species is crucial to improving conservation strategies. However, the pollinator observations in our study were made during the fruit-crop flowering season, and pollinator distribution may change due to mass-flowering or scarcity of flower resources at other times of year [33,60]. These may influence pollinator foraging behaviors which reflect pollination network structures [9,61,62]. We suggest further studies should investigate how distribution patterns of each pollinator group change between non-invaded and invaded areas before flowering and during flowering periods. Further research will be needed to investigate the influences of agrochemical use within the surrounding landscape on pollinator communities and explore the combined impacts with the effects of invasive plants.

## 4. Materials and Methods

### 4.1. Study Site and Species

The invasive plant praxelis (*Praxelis clematide**a* R.M. King and H. Rob.) is an annual/short-lived perennial herb belong to the Asteraceae family, and is a native to northern Argentina, southern Brazil, Bolivia, Paraguay and Peru [63]. It spreads rapidly due to diverse dispersal modes, it produces large numbers of seeds that are spread by wind, water, animals, and birds. In 2003, *P. clematidea* was first reported in Thailand in orchards and rubber plantations [64]. It has been reported as an invasive weed in many countries including Australia [65], USA [66], China [67], and countries in Southeast Asia [63]. According to their pollination syndromes, this kind of flower may be pollinated by bees, butterflies, and flies [63].

The study area is in the Northeastern part of Thailand, we selected smallholder orchards within the province Nakhon Ratchasima (14°58.5′ N 102°6′ E, Figure 6). The study area is located at the western edge of Khorat Plateau in Northeastern part of Thailand, average temperature ranges from 22.8 °C to 42.2 °C. The reference land-use map was obtained from the Land Development Department of Thailand (LDD) [68]. The study region is characterized by a highly agricultural landscape dominated by rice paddy-fields, sugarcane, cassava, and others crop plantations, also adjacent to two important national parks; including the edge of Khao Yai national park (2165.55 km^2^) in the west and Thap Lan national park (2245.88 km^2^) in the south [69]. Smallholder orchards are common in Northeastern Thailand as traditional farming systems for households and local markets [70,71]. The most common plants in smallholder orchards are crops, herbs, and flowers, with the crops consisting of plants such as mangoes (*Mangifera indica* L.), longans (*Dimocarpus longan* Lour. var. *longan*), tamarinds (*Tamarindus indica* L.), limes (*Citrus aurantifolia* (Christm.) Swingle [72]), papayas (*Carica papaya* L.), and Thai eggplant (*Solanum incanum* L.).

The study orchards ranged in size from 0.16 to 0.48 ha and every orchard had been managed for over three years. We focused on two smallholder orchard types: (1) invaded by *P. clematidea* and (2) non-invaded, nine sites were chosen for each orchard type. We calculated the proportion of three land-use types, i.e., agricultural land, forest patches, and urban areas within a 3 km radius from the center of each study site (Figure 6), due to the maximum foraging distances of 100–300 m for small bee species and up to 1100 m for large species [10,73]. All study sites were chosen based on similar plant communities, which consisted of at least ten flowering plant species and contained longan and mango trees. For invaded study sites, we selected the sites with *P. clematidea* covering a minimum of 20% of the ground cover, in order to standardize the distribution of invasive plants.

### 4.2. Sampling Design and Data Collection

At each study site, we set up a 50 × 50 m^2^ plot in the center of each orchard. We recorded and counted all flowering plant species in each plot as a measure of plant diversity only in the first time. We conducted observations of flowering abundance and pollinators from January to May 2021 (four times per site, as flowering started in February in some sites). The flower abundance and pollinator observation were sampled at the same time. To count the number of flowers, we counted each composite inflorescence as a flower, this allowed us to use consistent terminology and facilitate calculate meaningful metrics of floral density [74]. Plants were identified to species or genera based on the plant databases of the Botanical Garden Organization of Thailand [75] and Thai plant names [72].

In each plot, pollinator observations were conducted, as most flowering plants in study sites including *P. clematidea* show diurnal floral openings and nocturnal pollination is likely unaffected [34,63], survey took place between 08:00 and 11:00 according to the optimal time of day for insect activity [76,77] in favorable weather conditions (sunny and without rain and temperature ranging from 25 °C to 38 °C [52,78]). We recorded species and counted the number of pollinators when they visited the reproductive parts of flowers. We set four different quadrats in each plot then recorded the visitation for 15 min within a 30 × 30 cm^2^ plot that was placed at the location with the highest flower abundance of each quadrat. We set up five 1 × 1 m^2^ subplots to cover the flowering area of ground flora to observe flower-visitors for a period of 15 min per subplot. Each quadrat and subplot was set-up to cover the most flowering plants per visit to maximize focal flower-visit density. Pollinators were identified to order level in the field and captured pollinators during a further 15 min period for identification in the laboratory. Although identifying insects to species would have been ideal, the difficulty of identifying pollinators to the species level under field conditions prompted us to sort pollinators into morphotype level, as the accepted approach recommended by Kremen et al. (2002) [79]. We then categorized pollinators into four groups: (1) Hymenoptera (including honeybees, i.e., *Apis florea* and *A. cerana*; wild bees, i.e., stingless bees and other wild bees; wasps; and ants), (2) Lepidoptera (butterflies), (3) Diptera (files), and (4) other arthropods.

### 4.3. Pollination Network Construction and Analysis

All network metrics of the pollination network structure across season were visualized using the bipartite package in R software 3.6.3 [80]. Each network comprised a matrix A × P where A represent of pollinators and P represents flowering plants, and the cell values represent the number of visits made by pollinator species ‘A’ to plant species ‘P’ [81]. Although this metric is sensitive to sampling thoroughness, our pair networks utilize equal sampling effort making relative comparisons meaningful. To avoid statistical pseudo-replication, we pooled all observations data by study system (invaded and non-invaded). We analyzed two networks representing an independent pair of networks from two study systems, a network invaded by *P. clematidea* and a network without *P. clematidea*. Further, we also constructed a network of all study sites using pooled data from all study sites. For each invaded and non-invaded network, we calculated connectivity, interaction evenness based on Tylianakis et al. (2007)’s method [82], specialization (H′_2_) [83], and Shannon diversity of interactions [10] using network-level statistics in the bipartite package of R software [83]. Using group-level statistics in the bipartite package in R, the mean number of shared partners for the lower level (plant species) and higher level (pollinator species) were calculated for each study system network [83].

The ecological community networks can be considered assemblies of multiple species [25,84,85]. Subsets of highly linked nodes have many topological modules [86] which are important indicators of network complexity. It has been widely used to analyze modularity [25,26,87,88] and we believed that the modular structure of networks plays a critical role in their functionality [89]. To calculate modularity in bipartite network, we used the *QuanBiMo* algorithm (*Q*) provided by Dorman and Strauss 2014 [26] which is the default algorithm used the *co**mputeModules* function in bipartite package in R software [83]. We then used *nullmodel* function (100 randomizations, that were generated by vaznull method) to convert *Q* to a *z*-score, the value which above an approximately 2 are considered significant modular [26,87,88]. We calculated participation coefficients or *c* value (the among-module, measuring how connected a species is to all modules) and within-module degree or *z* value (standardized number of links to other species in the same module) for each pollinator species within the network to identify which species played the most important role in the network and investigate the role of invasive plant *P. clematidea* in the invaded network.

We sorted all species into peripherals (termed specialists), connectors, module hubs, and network hubs (three latter are termed generalists), used the critical *z* values of 2.5 and *c* value of 0.625 defined the role of species in the network [25,85]. Species with a low *c* ≤ 0.625 and low *z* ≤ 2.5 were peripheral species or specialists, they had only a few links and almost always only to species within their module (these even had a *c* = 0 which had no links at all outside their own module). Connector species linking several modules at a high *c* > 0.625 and low *z* ≤ 2.5, which glues modules together. Module hubs species have a low *c* ≤ 0.625 and high *z* > 2.5, highly connected species linked to many species within their own module. Network hubs or super generalist species which have both a high *c* > 0.625 and a high *z* > 2.5, acting as both connectors and module hubs and is thus important to the coherence of both the network and its own module [25,90,91,92].

### 4.4. Statistical Analysis

All analyses were conducted in R software 3.6.3 [80]. For each invasion state (invaded and non-invaded), we calculated the mean pollinator richness, pollinator abundance, and visitation rate. The pollinator diversity (Shannon–Wiener index) at each study site was calculated using vegan package version 2.6.2 in R [93]. To test the normality of all variables, we used the graphics package in R performed histogram and using Shapiro–Wilk normality test to make the assumption that datasets are normally distributed [80]. To assess differences in plant diversity, plant richness (all species in the plot), plant interacted species (which are observed the interactions with pollinators), pollinator richness, pollinator abundance, and visitation rate between study system, we used independent *t*-test as data were normal distributed, and differences in flower abundance and pollinator diversity between study system, we assessed with Wilcoxon rank sum test as data were non-normal distributed. The differences in visitation rate of Hymenopterans between study system was assessed using independent *t*-test. The differences in visitation rate of Lepidoptera, Diptera, and other arthropods between study system were assessed by Wilcoxon rank sum test. To compare the composition of pollinator species between invasion states, we used the non-metric multidimensional scaling (NMDS) with the *metaMDS* function of vegan package in R, based on Bray–Curtis distance and two dimensions. The significance of fitted vectors was assessed using permutation tests with 999 random permutations of data and demonstrated only significant variables (*p* ≤ 0.05). The differences in connectivity, interaction evenness, Shannon’s diversity of interactions, specialization (H′_2_), and mean number of shared partners for the lower level (plants) were assessed using Wilcoxon rank sum test, the mean number of shared partners higher level (pollinators) using independent *t*-test. We then compared *z* and *c* values between pollinator groups in each network system (invaded network and non-invaded network) using Wilcoxon rank sum test.

To test the effect of the proportion of landscape surrounding each study sites on pollinator community, firstly, a probability distribution that best fits the response variables was identified. Generalized linear models (GLMs) were constructed with the *glm* function of *stats* package in R [80]. We verified that assumption of normality and heteroscedasticity were met and none of the GLM was overdispersed. The spatial autocorrelation was checked by using *acf* function of *stats* package in R [80] and no autocorrelation was found in any model. Generalized linear models were fitted with Poisson distribution and log link to test the effect of landscape proportions surrounding each study site (agriculture land, forest, and urban area within a 3 km radius) on the following response variables: log link used for total pollinator richness and Hymenoptera richness and Poisson used for Diptera richness and Lepidoptera richness. The Gaussian distribution was used to determine the effect of landscape proportions surrounding each study site on following response variables: total pollinator visitation rate, each three pollinator groups visitation rate (i.e., Hymenoptera, Diptera, and Lepidoptera). Invasion state (invaded and non-invaded) was treated as fixed effect, and the plant rarefied richness and flower abundance also included as fixed effect. The proportion of agricultural land, forest, and urban area within a 3 km radius were included as explanatory variables. The interactions between explanatory variables that contribute at least marginally to the model (*p* < 0.10) were also added. To determine the best model, the GLMs with lowest Akaike’s information criteria (AIC) was selected (all model result is found in Appendix A). Regression between response variables and proportion of each landscape type within a 3 km radius were plotted with the corresponding confidence interval using ggplot2 package in R [80].

## 5. Conclusions

In summary, our study confirms that the majority of pollinators differed between invaded (*A. cerana* with 26.87%) and non-invaded (dipterans 20.65%) orchards (with 33% species overlapping). This study revealed the response of bees and flies to the invasive flower had an effect on pollinator community composition, which may promote generalist pollinators within study sites. The cultivated habitats at the landscape level influences functioning for both invaded and non-invaded communities, which is similar to results found in other studies which have demonstrated the landscape level impacts this species can have [94,95]. Moreover, the interaction between landscape proportions and plant community, such as plant richness and flower abundance, influenced the pollinator community. Contrary to our expectations, the invasive plant *P. clematidea* not impact the network structure directly, but its presence did alter the pollinator community. Because the peripheral species *P. clematidea* were visited by a lower diversity of pollinators, there was no shift in the specialization of plant–pollinator interactions within the invaded network, which is slightly different to other studies in parts of the region where changes were noted [96]. Consistent with our expectations, urbanization had a strong negative influence on pollinator communities while the presence of invasive plant increased pollinator richness. However, the proportion of agriculture and forest landscapes surrounding smallholder orchards was also associated with an increase in pollinator richness, which was reflected in the plant–pollinators interactions in the networks. The consequences of these changes on species interactions and the reproductive success of flowering plants, such as crop species, still requires further research regarding the long-term impacts of invasive plants on pollinator community dynamics.

## Figures and Tables

**Figure 1 plants-11-01976-f001:**
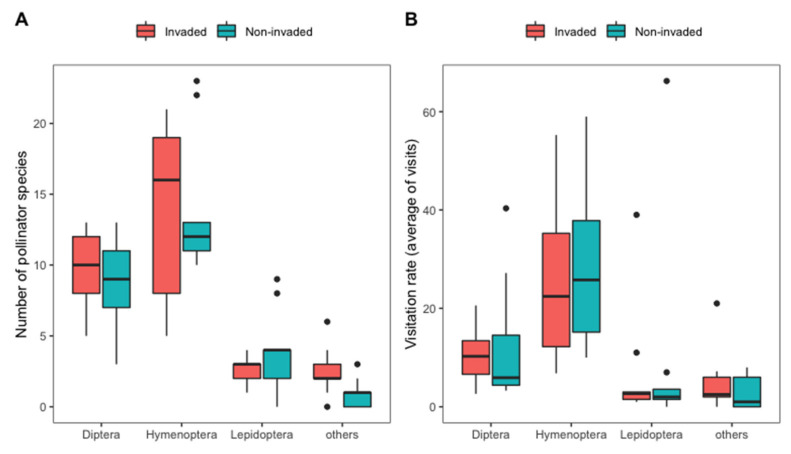
Pollinator richness (**A**) and visitation rates (**B**) from four insect groups (Diptera, Hymenoptera, Lepidoptera, and other arthropods) within the invaded and non-invaded sites.

**Figure 2 plants-11-01976-f002:**
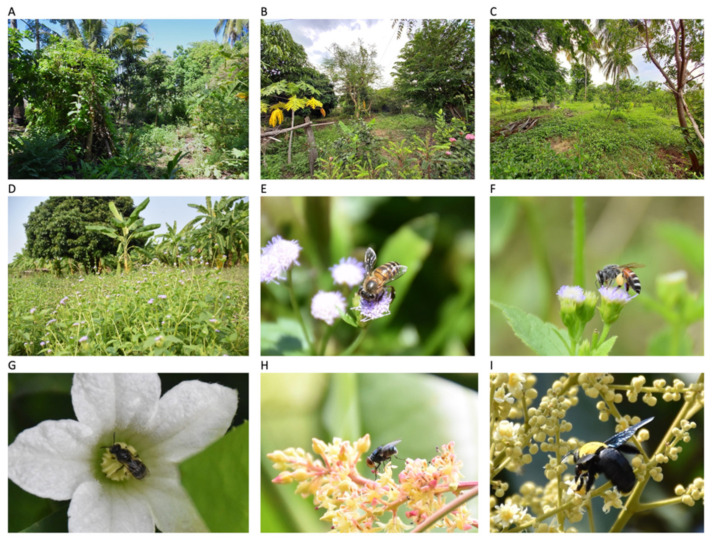
Example of the focal smallholder orchards (**A**–**C**), area invaded by *Praxelis clematide**a* (**D**), and pollinators; *Apis cerana* (**E**), *Apis florea* (**F**), *Halictid* sp. (**G**), *Lucilia* sp. (**H**), and *Xylocopa aestuans* (**I**).

**Figure 3 plants-11-01976-f003:**
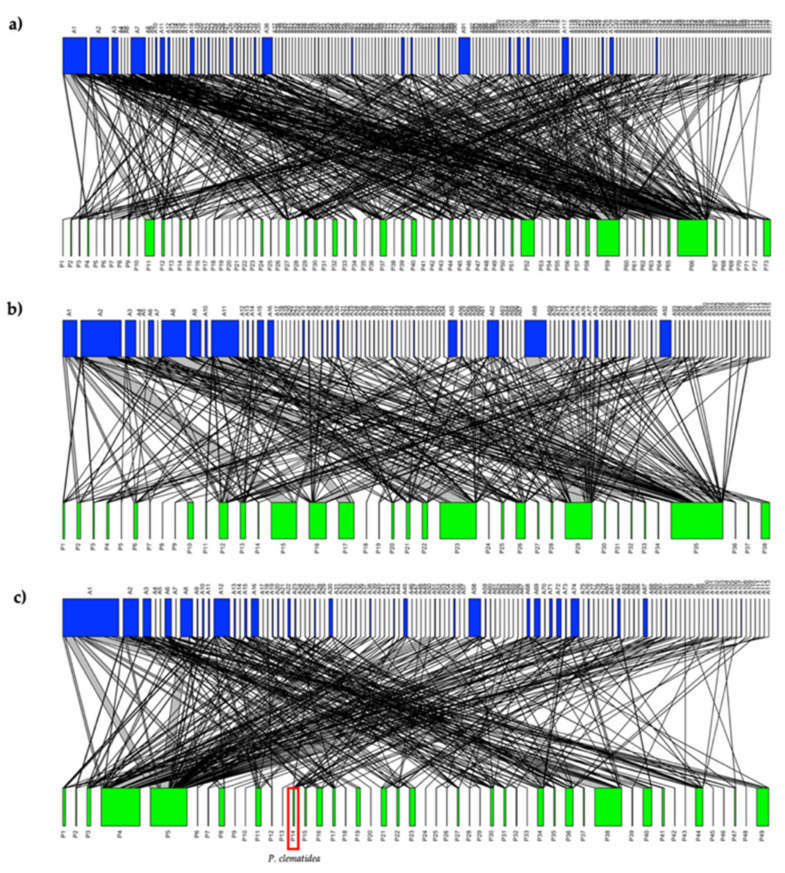
Pollination network showing the interactions between pollinators (top blue bars) and plants (bottom green bars), (the networks were combined for all sites and morphospecies for illustrative purposes); (**a**) network of all study sites (*n* = 18), (**b**) non-invaded network (*n* = 9), and (**c**) invaded network (*n* = 9). The width of the links is proportional to the number of interactions observed (the list of plants and pollinators species in these networks is found in Appendix A). Size and complexity should be examined rather than species names (other than Praxelis-which is shown in red).

**Figure 4 plants-11-01976-f004:**
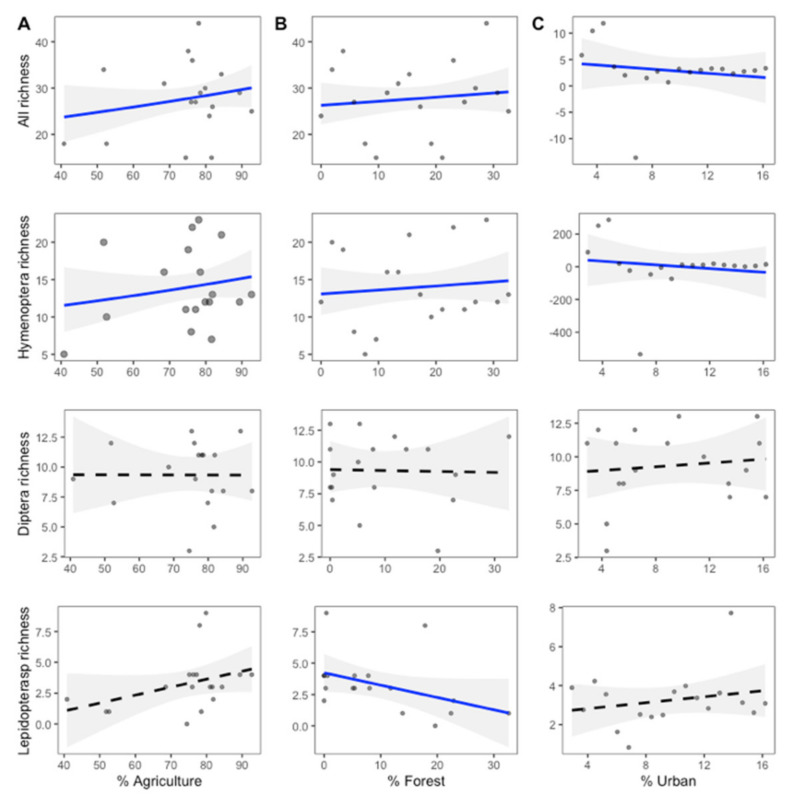
Pollinator richness in relation to the percentage of (**A**) agriculture, (**B**) forest, and (**C**) urban landscape within a 3 km radius from each study site. All regressions are plotted with 95% corresponding confidence intervals (scales have been optimized for each set of analysis and thus vary between plots). Solid blue lines indicate significant associations (*p* < 0.05), whereas dashed lines indicate non-significant relationships (*p* > 0.05). For statistics, see Appendix A.

**Figure 5 plants-11-01976-f005:**
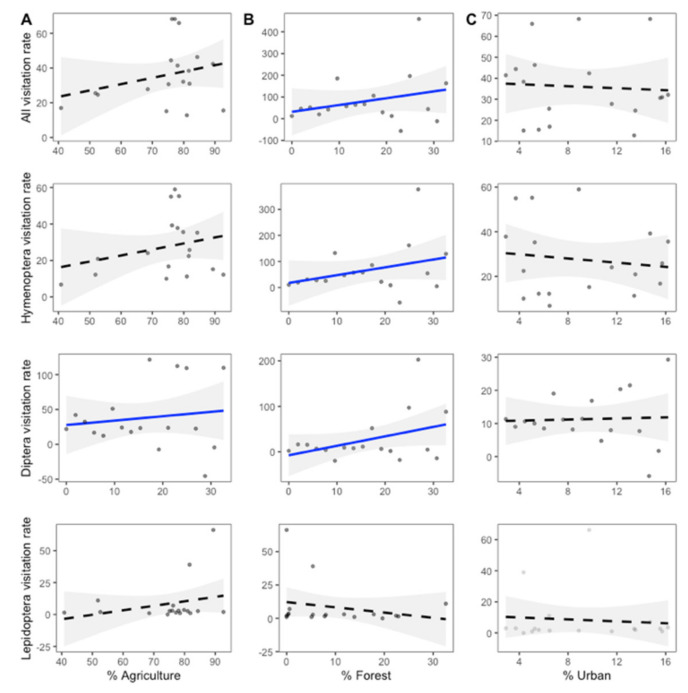
Visitation rates in relation to the percentage of (**A**) agriculture, (**B**) forest, and (**C**) urban landscape within a 3 km radius from each study site. All regressions are plotted with 95% corresponding confidented intervals (scales have been optimized for each set of analysis and therefore vary between plots). Solid blue lines indicate significant associations (*p* < 0.05), whereas dashed lines indicate a non-significant relationship (*p* > 0.05). For statistics, see Appendix A.

**Figure 6 plants-11-01976-f006:**
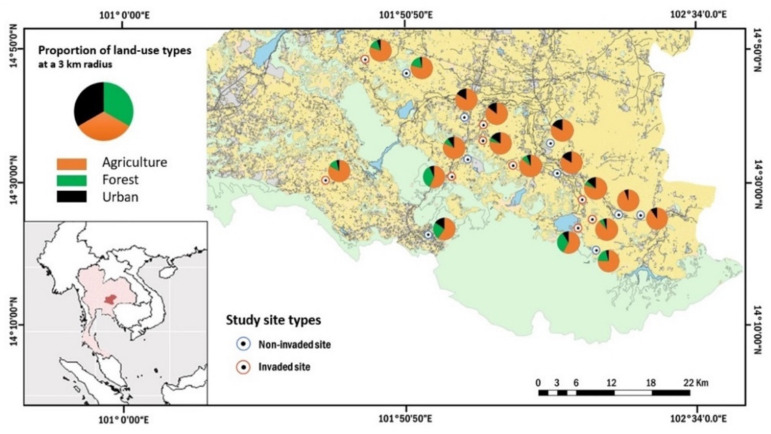
Locations of focal smallholder orchards (*n* = 18) in six districts of Nakhon Ratchasima province, Thailand. Pie-charts show the proportion of each landscape type in a 3 km radius around study sites, non-invaded sites are shown as blue circles and invaded sites as red circles.

**Table 1 plants-11-01976-t001:** Mean ± SD value of plant and pollinator community parameters. To assess differences in plant diversity, plant richness (all species in the plot), plant interacting species, pollinator richness, pollinator abundance, and visitation rate between study systems, we used an independent *t*-test. For the differences in flower abundance and pollinator diversity between invasion states, we employed Wilcoxon rank sum tests. The differences in Hymenopteran visitation rate between study systems was assessed using an independent *t*-test. The differences in the visitation rate of Lepidoptera, Diptera, and other arthropods between study systems were assessed with Wilcoxon rank sum test. ^NS^
*p* > 0.05.

Community	Variable	Invaded	Non-Invaded	*p*-Value
Plant	Diversity	3.38 ± 0.20	3.38 ± 0.15	0.98 ^NS^
	Richness	29.89 ± 5.97	29.78 ± 4.49	0.97 ^NS^
	Interacted species	8.33 ± 2.92	8.44 ± 3.05	0.94 ^NS^
	Flower abundance	2412.75 ± 1121.46	2839.60 ± 1944	0.86 ^NS^
Pollinator	Diversity	3.28 ± 0.31	3.28 ± 0.33	0.83 ^NS^
	Richness	27.67 ± 7.55	27.78 ± 8.71	0.98 ^NS^
	Pollinator abundance	514.67 ± 282.86	590 ± 399.6	0.65 ^NS^
	Visitation rate			
	All groups	34.31 ± 16.44	37.63 ± 19.83	0.70 ^NS^
	Hymenoptera	26.55 ± 18.23	28.41 ± 15.94	0.82 ^NS^
	Lepidoptera	7.18 ± 12.32	9.59 ± 21.34	0.93 ^NS^
	Diptera	11.12 ± 6.26	12.34 ± 13.04	0.60 ^NS^
	Other arthropods	4.86 ± 6.40	2.61 ± 3.48	0.26 ^NS^

**Table 2 plants-11-01976-t002:** Mean ± SD value of the network structure parameters. Connectivity, interaction evenness, Shannon’s diversity of interactions, specialization (H′_2_), and mean number of shared partners for the lower level (plants) were analyzed using Wilcoxon rank sum test. Mean number of shared partners higher level (pollinators) was assessed using a *t*-test. ^NS^
*p* > 0.05.

Parameters	Invaded Network	Non-Invaded Network	*p*-Value
Connectance	0.19 ± 0.06	0.19 ± 0.05	0.73 ^NS^
Interaction evenness	0.51 ± 0.07	0.49 ± 0.06	0.44 ^NS^
Shannon’s diversity of interactions	2.76 ± 0.54	2.60 ± 0.41	0.67 ^NS^
Specialization (H′_2_)	0.70 ± 0.14	0.77 ± 0.12	0.27 ^NS^
Lower level shared partners (plants)	0.79 ± 0.53	0.71 ± 0.37	0.79 ^NS^
Higher level shared partners (pollinators)	0.44 ± 0.12	0.47 ± 0.17	0.63 ^NS^

## Data Availability

Data are provided as private-for-peer review at https://figshare.com/s/c094d1e8f53a0adcf254 (accessed on 1 January 2020). and the data will be permanently archived if the paper is accepted for publication.

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
