# Peer review of "Effect of Landscape Composition and Invasive Plants on Pollination Networks of Smallholder Orchards in Northeastern Thailand"

_plants, 2022, doi:10.3390/plants11151976_

Round 1

Reviewer 1 Report

Dear Authors,

The submitted manuscript titled „Effect of landscape composition and invasive plant on pollination networks of smallholder orchards in Northeastern Thailand” presents very interesting and valuable results, which might interest the international audience. The manuscript is generalny well-written, however, I have some suggestion of improvements, which I have listed below:

1.            The choice of study species P. clematidea should be stronger justify  in chapter Introduction.

2.            In my opinion the Figure with observation design presented in lines 428-445 should be very useful.

3.            The proposed future directions of investigations on P. clematidea should be presented.

4.            I suggest to look into the  following papers, which would be useful in providing eventual improvements in manuscript:

·         Wei H, Xu J, Quan G, Zhang J, Qin Z. Effects of Praxelis clematidea invasion on soil nitrogen fractions and transformation rates in a tropical savanna. Environ Sci Pollut Res Int. 2017,24(4):3654-3663. doi: 10.1007/s11356-016-8127-6

·         Huang XR, Pang SL, Shen WH, Hou YR, He F. Herb diversity and its affecting factors of community invaded by Praxelis clematidea in karst mountainous area of Guangxi Province, China. The Journal of Applied Ecology. 2016 Mar;27(3):815-821. DOI: 10.13287/j.1001-9332.201603.016.

·         Ya Wang, Weiqian Wang, Qinke Wang, Xiaoxia Li, Yan Liu, Qiaoqiao Huang. Effects of soil nutrients on reproductive traits of invasive and native annual Asteraceae plants[J]. Biodiv Sci, 2021, 29(1): 1-9.

Author Response

Dear Authors,

The submitted manuscript titled „Effect of landscape composition and invasive plant on pollination networks of smallholder orchards in Northeastern Thailand” presents very interesting and valuable results, which might interest the international audience. The manuscript is generalny well-written, however, I have some suggestion of improvements, which I have listed below:

Response: Thank you for the useful comments, we have worked through the text to revise as suggested

  1. The choice of study species P. clematidea should be stronger justify  in chapter Introduction.

Response: Thank you for the comment. We have added the information about its threats. Line 82-87

  1. In my opinion the Figure with observation design presented in lines 428-445 should be very useful.

Response: Thank you for the suggestions.

  1. The proposed future directions of investigations on P. clematidea should be presented.

Response: Thank you very much for the suggestions. We added the suggestions for future studies to investigate the effective competition for pollination between the invasive plant P. clematidea and native plants. Line 403-406

  1. I suggest to look into the  following papers, which would be useful in providing eventual improvements in manuscript:

  Wei H, Xu J, Quan G, Zhang J, Qin Z. Effects of Praxelis clematidea invasion on soil nitrogen fractions and transformation rates in a tropical savanna. Environ Sci Pollut Res Int. 2017,24(4):3654-3663. doi: 10.1007/s11356-016-8127-6

  Huang XR, Pang SL, Shen WH, Hou YR, He F. Herb diversity and its affecting factors of community invaded by Praxelis clematidea in karst mountainous area of Guangxi Province, China. The Journal of Applied Ecology. 2016 Mar;27(3):815-821. DOI: 10.13287/j.1001-9332.201603.016.

Ya Wang, Weiqian Wang, Qinke Wang, Xiaoxia Li, Yan Liu, Qiaoqiao Huang. Effects of soil nutrients on reproductive traits of invasive and native annual Asteraceae plants[J]. Biodiv Sci, 2021, 29(1): 1-9.

Response: Thank you, added.

Reviewer 2 Report

Review of the manuscript "Effect of landscape composition and invasive plant on pollination networks of smallholder orchards in Northeastern Thailand" (Manuscript ID: plants-1834977 )

Several studies have been showed that invasive plants have serious consequences on plant-pollinator interactions throught the modification of native plant communities. Nevertheless, there are hardly any studies where investigated the effect of invasive plant on pollinator networks in smallholder orchards. I find the approach to the topic interesting and novel. The introduction is correct and wel written. However, the rest of the manuscript contains errors that need strong corrections.

I lack representative images of the examined areas/stands or communities (plants/pollinators). This could make the manuscript much more attractive and enjoyable (and brings the reader closer to the problem).

All figures, tables and even the text of the MS begin the discussion/description with the invaded stands, followed by the non-invaded stands. This should be reversed everywhere.

The figure captions are too small (Fig 1, 3, 4 and 5), they can be at least 3x larger. It is also necessary to unify the fonts of the figures.

Figure 2 currently shows nothing. The captions are completely illegible. In addition, (as far as I can see..) these abbreviations are only resolved in the Supplement, perhaps, but there is no indication of this. Instead of abbreviations, it would be better to use species names and/or families (and possibly mark them with other colors instead of green and blue).

In the case of the Fig 3 and 4, scale of the charts is not uniform, this can mislead the reader.

The figures must be inserted in the appropriate place (after the first citationss in the main text cf.: instructions for authors/preparing figs, schemes and tables).

The Results section is too long and contains phrases like Discussion. It is necessary and sufficient to shorten this and eliminate the parts that do not belong here. The Materials and Methods should also be shortened, especially what belongs to subsections 4.3 and 4.4.

Other comments:

Line 28: 3 km, instead of 3km

L 78: It would be more appropriate to start the last paragraph of the Introduction here: “Here we focus on the modularity of …”

L 109: Odonata 1 ?? Can dragonflies (Odonata) be pollinators?

L 181-182: First write the name of the model, then indicate its abbreviation in parentheses, then you can use the abbreviation. Generalized Linear Model (GLM)

L 190: and ?? and what? The sentence is not finished.

L 268, 309, 362, 367 and 488: Remove highlighting of citations.

L 281: Missing space. “..habitat [6,11].” Instead of “..habitat[6,11].”

L 399-400: The scientific name of the species is written in italics (except for the word var.): Dimocarpus longan var. longan).

L 415: Possibly referred to as Pie-charts.

L 422: January to May: 5 times per site (this means 5 times, if we include May).

L 429: Ok. I’m just wondering: What effect might the invasive plant species have on nocturnal pollinators? It can be interesting.

Despite all of this, I consider the manuscript suitable for publication in this Journal if the correction of the above mentioned errors and omissions is carried out by the authors.

Author Response

Reviewer 2

Review of the manuscript "Effect of landscape composition and invasive plant on pollination networks of smallholder orchards in Northeastern Thailand" (Manuscript ID: plants-1834977 )

Several studies have been showed that invasive plants have serious consequences on plant-pollinator interactions throught the modification of native plant communities. Nevertheless, there are hardly any studies where investigated the effect of invasive plant on pollinator networks in smallholder orchards. I find the approach to the topic interesting and novel. The introduction is correct and wel written. However, the rest of the manuscript contains errors that need strong corrections.

 Response: Thank you, we are glad the paper is useful, and hope it enhances our knowledge of the impacts of invasive species from a community perspective

I lack representative images of the examined areas/stands or communities (plants/pollinators). This could make the manuscript much more attractive and enjoyable (and brings the reader closer to the problem).

Response: We have added study site images.

All figures, tables and even the text of the MS begin the discussion/description with the invaded stands, followed by the non-invaded stands. This should be reversed everywhere.

Response: This has not been changed, we feel an uninvaded baseline should be used first as any change is relative to the baseline and it provided a clear reflection of the impact of invasion.

The figure captions are too small (Fig 1, 3, 4 and 5), they can be at least 3x larger. It is also necessary to unify the fonts of the figures.

Response: The figures are provided in high resolution, so can be opened and zoomed in on.

Figure 2 currently shows nothing. The captions are completely illegible. In addition, (as far as I can see..) these abbreviations are only resolved in the Supplement, perhaps, but there is no indication of this. Instead of abbreviations, it would be better to use species names and/or families (and possibly mark them with other colors instead of green and blue).

Response: This is standard for a pollination network, the structural change is what matters in these images. The legend has been modified to facilitate interpretation to key patterns and shifts are easier to interpret

In the case of the Fig 3 and 4, scale of the charts is not uniform, this can mislead the reader.

Response: The legend under the figure has been changed to make the differences clear

The figures must be inserted in the appropriate place (after the first citationss in the main text cf.: instructions for authors/preparing figs, schemes and tables).

Response: The correct format was followed for the submission and figures are already in the correct places.

The Results section is too long and contains phrases like Discussion. It is necessary and sufficient to shorten this and eliminate the parts that do not belong here. The Materials and Methods should also be shortened, especially what belongs to subsections 4.3 and 4.4.

Response: We feel the length is appropriate to understand and replicate the study, so this has not been changed. We hope that this makes the paper more accessible, replicable and relevant to readers.

Other comments:

Line 28: 3 km, instead of 3km

Response: km is the unit, spacing is optional here

L 78: It would be more appropriate to start the last paragraph of the Introduction here: “Here we focus on the modularity of …”

Response: This has been changed.

L 109: Odonata 1 ?? Can dragonflies (Odonata) be pollinators?

Response: we now use the phrase flower visitors as the phrase because pollination efficacy was not tested, and odonata are not tested further

L 181-182: First write the name of the model, then indicate its abbreviation in parentheses, then you can use the abbreviation. Generalized Linear Model (GLM)

Response: Changed as suggested. Line 196

L 190: and ?? and what? The sentence is not finished.

Response: Corrected. Line 205

L 268, 309, 362, 367 and 488: Remove highlighting of citations.

Response: Changed as suggested.

L 281: Missing space. “..habitat [6,11].” Instead of “..habitat[6,11].”

Response: Corrected. Line 317

L 399-400: The scientific name of the species is written in italics (except for the word var.): Dimocarpus longan var. longan).

Response: Corrected. Line 439

L 415: Possibly referred to as Pie-charts.

Response: Changed as suggested.

L 422: January to May: 5 times per site (this means 5 times, if we include May).

Response: We actually observed each site four times, as some sites started flowering later (February), corrected in text.

L 429: Ok. I’m just wondering: What effect might the invasive plant species have on nocturnal pollinators? It can be interesting.

 Response: noted as most flowering plants in study sites including P. clematidea show diurnal floral openings and nocturnal pollination is likely unaffected

Despite all of this, I consider the manuscript suitable for publication in this Journal if the correction of the above mentioned errors and omissions is carried out by the authors.

 Response: Thank you!

Round 2

Reviewer 2 Report

The authors have made the most necessary corrections. As I indicated, I consider the manuscript suitable for publication in this Journal.